# Light-tuned selective photosynthesis of azo- and azoxy-aromatics using graphitic $C_3N_4$

Yitao Dai[1,2], Chao Li[2], Yanbin Shen[2,5], Tingbin Lim[2], Jian Xu[2], Yongwang Li[2], Hans Niemantsverdriet[2,3], Flemming Besenbacher[1], Nina Lock[4] & Ren Su[2]

Solar-driven photocatalysis has attracted significant attention in water splitting, $CO_2$ reduction and organic synthesis. The syntheses of valuable azo- and azoxyaromatic dyes via selective photoreduction of nitroaromatic compounds have been realised using supported plasmonic metal nanoparticles at elevated temperatures ($\geq$90 °C); however, the high cost, low efficiency and poor selectivity of such catalyst systems at room temperature limit their application. Here we demonstrate that the inexpensive graphitic $C_3N_4$ is an efficient photocatalyst for selective syntheses of a series of azo- and azoxy-aromatic compounds from their corresponding nitroaromatics under either purple (410 nm) or blue light (450 nm) excitation. The high efficiency and high selectivity towards azo- and azoxy-aromatic compounds can be attributed to the weakly bound photogenerated surface adsorbed H-atoms and a favourable N-N coupling reaction. The results reveal financial and environmental potential of photocatalysis for mass production of valuable chemicals.

[1] Interdisciplinary Nanoscience Centre (iNANO), Aarhus University, Gustav Wieds Vej 14, DK-8000 Aarhus C, Denmark. [2] SynCat@Beijing, Synfuels China Technology Co. Ltd., Leyuan South Street II, No. 1, Yanqi Economic Development Zone C#, Huairou District, 101407 Beijing, China. [3] SynCat@DIFFER, 6336 HH Eindhoven, The Netherlands. [4] Carbon Dioxide Activation Center, Interdisciplinary Nanoscience Centre (iNANO) and Department of Chemistry, Aarhus University, Gustav Wieds Vej 14, DK-8000 Aarhus C, Denmark. [5] Present address: Suzhou Institute of Nano-Tech and Nano-Bionics (SINANO), No. 398 Ruoshui Road, Suzhou Industrial Park, Jiangsu Province, 215123 Suzhou, China. Correspondence and requests for materials should be addressed to N.L. (email: nlock@inano.au.dk) or to R.S. (email: suren@synfuelschina.com.cn)

zo- and azoxyaromatic compounds are very important and precious precursors for the dye-, electronic-, pigment-, and drug industries[1]. Traditional industrial syntheses of azo- and azoxy-aromatic compounds are realised by diazotisation (yields >70%). The process is costly and environmentally unfriendly, due to the formation of unstable intermediate diazo compounds and harsh process conditions (involving strict temperature control and the use of corrosive acids)[2,3]. In a more environmentally friendly approach, azo compounds can be obtained via aerobic oxidation of amines by heterogeneous catalysis, but this requires supported noble metal catalysts or expensive oxidants, which limits large-scale applications[4,5].

Heterogeneous photocatalysis shows great potential in driving several important reactions (i.e., water splitting, $CO_2$ reduction, and organic synthesis), and the technology has therefore attracted significant attention over the last decades[6–10]. Recent development of the graphitic $C_3N_4$ (g-$C_3N_4$) has further boosted the fundamental research and industrial potential of photocatalysis, as this inexpensive, metal-free material exhibits reasonable catalytic performance and stability under visible light irradiation conditions[11–15].

A number of studies have reported on the photoreduction of nitroaromatic compounds, either by inorganic semiconductor photocatalysts or by the supported plasmonic metal nanoparticles (NPs), as shown in (Table 1)[16–24]. The photocatalyst producing azo- or azoxy-compounds at low temperatures typically contains a precious metal. For example, a rhodium containing photocatalyst ($SiO_2$/CdS/Rh) converts nitrobenzene selectively to azoxybenzene under visible light irradiation[18]. However, in addition to the low efficiency and poor selectivity, the use of the unstable and toxic CdS photocatalyst makes this process unfavourable for industrial applications (Table 1, entry 1). Azobenzene can be successfully synthesised at room temperature (RT) by using a Au supported $TiO_2$ photocatalyst (Table 1, entry

2)[19]. A recent study has demonstrated that selective photosynthesis of azobenzene can also be realised at elevated temperature (90 °C), by using a cheaper metal (Cu) supported on graphene (Table 1, entry 3)[17,20]. However, since the nitrobenzene reduction can be initiated at 80 °C without any catalyst (poor activity and selectivity)[21], the use of an expensive graphene support in a reaction that is inefficient at RT, affects its commercial potential negatively. In addition, the supported plasmonic metal nanoparticle based photocatalysts (i.e., Au, Ag, and Cu) show a relatively poor efficiency for light absorption compared to their semiconductor counterpart, and the particle size of the metal would have to be controlled strictly to absorb photons with the desired energies[25]. Instead of azo- and azoxy-compounds, amines, i.e. the fully hydrogenated lower-value products, were reported as the main products in many studies when using conventional semiconductor photocatalysts such as $TiO_2$ and CdS (Table 1, entries 4–6)[22–24].

Up to now, it has been challenging to realise efficient photosynthesis of azo- and azoxy-aromatic compounds with controlled selectivity at RT without producing amines. Herein, we show that both azoxybenzene and azobenzene can be synthesised by photoreduction of nitrobenzene with high selectivity using the cheap graphitic $C_3N_4$ (g-$C_3N_4$, <0.1 \$ $g^{-1}$ on lab-scale) as photocatalyst at RT. The selectivity can be simply controlled by the irradiation wavelength. We estimate the turn over number (TON) to be ~8600 per surface active site (i.e., the two-coordinated N) over the reaction time, suggesting that the process is catalytic.

## Results

**Lab-scale nitrobenzene conversion.** As described in Table 1, the selective conversion of nitrobenzene to azobenzene and azoxybenzene can be controlled by the irradiation wavelength (450 and 410 nm, entry 7 and 8). Surprisingly, an amorphous graphitic $C_3N_4$ compound (A-g-$C_3N_4$) characterised by a larger specific

**Table 1 Possible products formed by photocatalytic reduction of nitroaromatic compounds and comparison of the conversion and selectivity of different photocatalysts**

| Entry | Photocatalysts | Cost (€ $g^{-1}$) | Radiation | Temperature (°C) | Conversion (%) | Selectivity (%) | | |
|---|---|---|---|---|---|---|---|---|
| | | | | | | Azoxy- | Azo- | Aniline |
| 1 | $SiO_2$/CdS/Rh[a] | 7.7 | 436 nm, 25 h | RT | 80 | 68 | 5 | 5 |
| 2 | Au/$TiO_2$[b] | 14.4 | >430 nm, 12 h | RT | 95 | 0 | 99 | N/A |
| 3 | Cu/Graphene[c] | 92.2 | >400 nm, 5 h | 90 | 98 | N/A | 98 | N/A |
| 4 | $TiO_2$/$N_3$/Pt[d] | 57.1 | 530 nm, 24 h | RT | >99 | 0 | 0 | 100 |
| 5 | Au/$TiO_2$-Ag[e] | 7.6 | >450 nm, 10 h | RT | 100 | 0 | 0 | 100 |
| 6 | $Ni_2$P/CdS[f] | 6.2 | >420 nm, 24 h | RT | 99 | 0 | 0 | >99 |
| 7 | g-$C_3N_4$[g] | 0.1 | 450 nm, 12 h | RT | 97 | 95 | 5 | 0 |
| 8 | g-$C_3N_4$[h] | 0.1 | 410 nm, 5 h | RT | 95 | 6 | 94 | 0 |
| 9 | A-g-$C_3N_4$[g] | 0.1 | 450 nm, 12 h | RT | 1 | 99 | 0 | <1 |

[a] High-pressure Hg lamp (0.007 W $cm^{-2}$)[18]
[b] 0.01 M KOH, 300 W Xe lamp (>430 nm, 0.01 W $cm^{-2}$)[19]
[c] 0.01 M KOH, 300 W Xe lamp (400–800 nm, 0.15 W $cm^{-2}$)[16]
[d] N3 dye on $TiO_2$ with Pt, triethanolamine (TEOA) as scavenger, 3 W 530 nm LED[26]
[e] 500 W Xe lamp (450–600 nm, 0.083 W $cm^{-2}$)[24]
[f] LED lamp (>420 nm, 30 × 3 W), $Na_2$S and $Na_2SO_3$ in water[23]
[g] 0.01 M KOH, 450 nm LED (440–460 nm, 0.03 W $cm^{-2}$)
[h] 0.01 M KOH, 410 nm LED (400–420 nm, 0.03 W $cm^{-2}$)

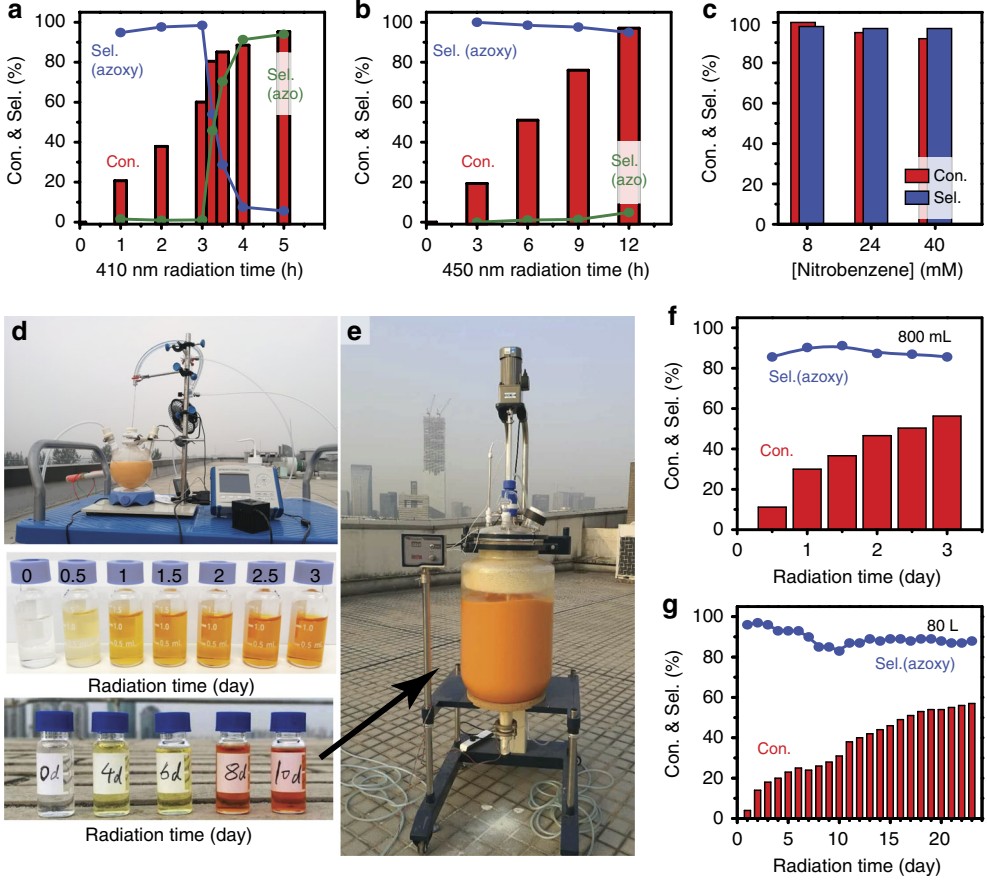

**Fig. 1** Photocatalytic performance. **a**, **b** Photoconversion (Con.) and selectivity (Sel.) of nitrobenzene to azobenzene and azoxybenzene using g-C$_3$N$_4$ under 410 and 450 nm irradiation (lab-scale). **c** Photoconversion of nitrobenzene to azoxybenzene using g-C$_3$N$_4$ at different starting concentrations under 450 nm irradiation. Reaction conditions: 8 mM nitrobenzene, 10 mM KOH, and 50 mg catalyst in 10 mL isopropanol under N$_2$ atmosphere. **d**, **e** Up-scaled reactions with volumes of 800 mL and 80 L, respectively. The nitrobenzene photoconversion tests were performed under solar radiation in Beijing (30/9/2016-2/10/2016, 25–30 °C, 800 mL) and Shenzhen (18/12/2016-5/1/2017, 25–30 °C, 80 L). The starting concentration of nitrobenzene was 8 mM. **f**, **g** Conversion and selectivity for the scaled-up tests

surface area (112 m$^2$ g$^{-1}$) compared to that of g-C$_3$N$_4$ (49 m$^2$ g$^{-1}$) shows a negligible performance in photocatalytic nitrobenzene reduction under similar reaction conditions (Table 1, entry 9).

Under 410 nm irradiation, azobenzene is gradually formed in a two-step reduction reaction; firstly, nitrobenzene is converted to azoxybenzene, which is then subsequently reduced to azobenzene (Fig. 1a). In contrast, azoxybenzene has been found to be the main product throughout the 450 nm irradiation course (Fig. 1b). Under both irradiation conditions, the nitrobenzene reduction can reach high conversion (>95%) with high selectivity (>94%) without the formation of aniline. We estimate that the quantum efficiencies (QE) are 1.4 and 0.4% for the synthesis of azobenzene and azoxybenzene under 410 and 450 nm irradiation, respectively (Supplementary Equations 1 and 2). It should be noted that such QEs are relatively high considering that the reactions involve 6e$^-$ and 8e$^-$ charge transfer processes, respectively, to achieve azoxy- and azobenzene. We also observed that the selectivity to aniline remains very low (≤5%) even under UV irradiation until most nitrobenzene molecules (>82%) had been converted (Supplementary Table 1).

**Up-scaling.** The photoconversion of nitrobenzene into the value-added azoxybenzene by using the g-C$_3$N$_4$ photocatalyst shows a huge potential for scaling-up to industrial level, both in terms of the nitrobenzene starting concentration and the reaction volume.

As shown in Fig. 1c, the high selectivity and conversion observed at low nitrobenzene concentration (8 mM) are maintained at higher nitrobenzene concentrations (24 and 40 mM) under identical reaction conditions. We observed that prolonged irradiation times are required to reach full conversion (12, 60, and 168 h, for 8, 24, and 40 mM of nitrobenzene, respectively), which may be caused by the reduction of light transmission to the photocatalyst due to the increased light absorption of azoxybenzene.

Figure 1d and e depict the reactions scaled-up in volume (to 0.8 and 80 L corresponding to an 80 and 8000-fold volume increase compared to the lab-scale reactions) performed under solar irradiation (i.e., without the use of artificial light sources). The yellow catalyst suspension gradually turned orange, indicating the formation of azo- and azoxybenzene. The colour change of the liquid caused by the formation of azo- and azoxybenzene was even more obvious after centrifugation. Gas chromatography (GC) analysis reveals that the high selectivity (~90%) towards azoxybenzene was achieved for both up-scaled tests (Fig. 1f and g). The solar irradiation favours the formation of the same compound as that formed under blue light irradiation (450 nm, azoxybenzene), which suggests that the reduction of nitrobenzene to azoxybenzene is more favourable than that of the conversion of azoxybenzene to azobenzene. The formation of azobenzene will only take place when nitrobenzene has been fully converted to azoxybenzene. We estimate the QE of the 8000-fold

**Table 2 Photocatalytic reduction of various nitroaromatic compounds to azoxy- (I) and azo- (II) aromatic compounds using the g-C$_3$N$_4$ photocatalyst**

| Entry | R | Light (nm) | Time (h) | Conversion (%) | Selectivity (%) | | Ratio | |
|---|---|---|---|---|---|---|---|---|
| | | | | | I + II | Amine | I | II |
| 1 | p-Cl | 410 | 8 | 100 | 100 | 0 | 2 | 98 |
| 2 | | 450 | 12 | 100 | 100 | 0 | 92 | 8 |
| 3 | p-Br | 410 | 20 | 95 | 100 | 0 | 5 | 95 |
| 4 | | 450 | 14 | 92 | 90 | 10 | 99 | 1 |
| 5 | p-I | 410[a] | 38 | 98 | 100 | 0 | 3 | 97 |
| 6 | | 450[b] | 60 | 100 | 96 | 4 | 94 | 6 |
| 7 | o-Cl | 410[c] | 35 | 100 | 100 | 0 | 4 | 96 |
| 8 | | 450[c] | 36 | 95 | 100 | 0 | 92 | 8 |
| 9 | m-Cl | 410 | 13 | 100 | 100 | 0 | 8 | 92 |
| 10 | | 450[d] | 12 | 100 | 90 | 10 | 63 | 37 |
| 11 | p-CH$_3$ | 410[b] | 20 | 100 | 91 | 9 | 7 | 93 |
| 12 | | 450[e] | 12 | 96 | 99 | 1 | 97 | 3 |
| 13 | p-OCH$_3$ | 410[f] | 20 | 100 | 100 | 0 | 0 | 100 |
| 14 | | 450[g] | 24 | 90 | 100 | 0 | 93 | 7 |
| 15 | p-CF$_3$ | 410 | 14 | 90 | 94 | 6 | 5 | 95 |
| 16 | | 450[g] | 12 | 100 | 100 | 0 | 98 | 2 |
| 17 | p-Cl, m-F | 410 | 12 | 60 | 100 | 0 | 0 | 100 |
| 18 | | 450 | 12 | 80 | 100 | 0 | 95 | 5 |
| 19 | p-Cl, m-CF$_3$ | 410[b] | 30 | 100 | 100 | 0 | 0 | 100 |
| 20 | | 450[g] | 60 | 91 | 100 | 0 | 90 | 10 |
| 21 | p-CH$_3$, m-CH$_3$ | 410[h] | 24 | 100 | 100 | 0 | 3 | 97 |
| 22 | | 450[e] | 20 | 93 | 100 | 0 | 99 | 1 |
| 23 | p-CF$_3$, m-CF$_3$ | 410[b] | 22 | 100 | 100 | 0 | 0 | 100 |
| 24 | | 450[i] | 38 | 100 | 100 | 0 | 90 | 10 |

General reaction conditions: 8 mM substrates, 30 mW cm$^{-2}$ light intensity, 10 mM KOH, and 50 mg catalyst in 10 mL isopropanol under deaerated condition
[a]20 mW cm$^{-2}$, 4 mM p-iodo nitrobenzene
[b]20 mW cm$^{-2}$
[c]20 mW cm$^{-2}$, 20 mM KOH
[d]30 mW cm$^{-2}$, 20 mM KOH
[e]30 mW cm$^{-2}$, 40 mM KOH
[f]10 mW cm$^{-2}$, 40 mM KOH
[g]30 mW cm$^{-2}$
[h]20 mW cm$^{-2}$, 40 mM KOH
[i]10 mW cm$^{-2}$, 20 mM KOH

scaled-up test to be 1.9% (see Fig. 1f, g and Supplementary Figs 6 and 7). The higher QE observed for the scaled-up test than that of the lab-scale experiment can be associated to with utilisation of the UV light in the solar spectrum (300–400 nm). We also observed that the conversion increased steadily with irradiation time in both up-scaled experiments with a similar rate (Fig. 1f and g). This suggests a strong photostability of the g-C$_3$N$_4$ and that scaling-up only has a negligible effect on the performance, which most likely allows the reactions to be even further scaled-up for industrial applications. We notice a relatively low conversion (~ 60%) for both scaled-up reactions, which may be caused by the accumulation of azoxybenzene that blocks the light absorption of the g-C$_3$N$_4$ photocatalyst. Further optimisation of the reactor design could possibly solve this technique issue (i.e., by using a flow cell or shorter light path length).

**Photoconversion of other nitroaromatic compounds.** By tuning the irradiation wavelength of the light sources, we have f`urther demonstrated the versatility of the g-C$_3$N$_4$ catalyst for controlled photoconversion of various nitroaromatic compounds into their corresponding azoxy- or azo-compounds (Table 2). The compounds p,p′-dichloro-, p,p′-dibromo-, and p,p′-iodo- azo- and azoxybenzene can be synthesised with excellent conversion and selectivity from the p-chloro-, p-bromo-, and p-iodo nitrobenzene, respectively (Table 2, entries 1–6). Likewise, the o,o′-dichloro- and m,m′-dichloro- substituted azo- and azoxybenzene can also be selectively synthesised from the corresponding o-chloro nitrobenzene and m-chloro nitrobenzene (Table 2, entries 7–10). Furthermore, the photoconversion of p-nitrotoluene, p-nitroanisole and p-nitrobenzotrifluoride into their corresponding azo- and azoxybenzene can also be achieved with very high conversion and selectivity in a controlled manner (Table 2, entries 11–16). We have also examined the photoconversion of four di-substituted nitrobenzene precursors (p-Cl, m-F; p-Cl, m-CF$_3$; p-CH$_3$, m-CH$_3$; and p-CF$_3$, m-CF$_3$) (Table 2, entry 17–24). All of these di-substituted nitroaromatic compounds can be converted into their corresponding azo- and azoxy-aromatic products with high selectivity, indicating that more complicated azoxy- and azo-aromatic compounds can be synthesised via this approach. In addition, the photosynthesis of asymmetric azobenzenes can also be realised with a reasonable selectivity (Supplementary Table 6).

**Reaction mechanism.** We propose that the excellent performance of the g-C$_3$N$_4$ photocatalyst for nitrobenzene photoreductions can be associated with the efficient utilisation of photogenerated

electrons for the reduction reaction. This has been revealed by mechanistic studies of the reduction of $O_2$ by using in-situ mass spectrometry (MS, Fig. 2a). Under deaerated conditions, the remaining oxygen that was dissolved in the catalyst-isopropanol suspension was reduced rapidly under 450 nm LED irradiation

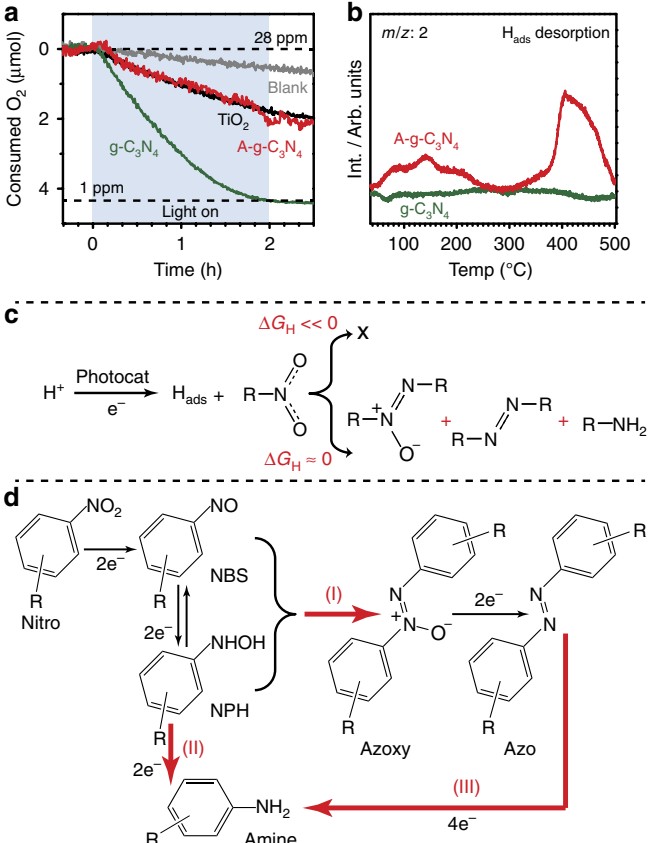

**Fig. 2** Reaction mechanism analysis. **a** The consumption of dissolved $O_2$ ($O_2$ reduction and isopropanol oxidation) under irradiation using various photocatalysts determined by in-situ MS. A 365 nm LED was used for $TiO_2$ and a 450 nm LED was used for all other tests. **b** Post-mortem TPD spectra revealing the desorption of $H_{ads}$ from the g-$C_3N_4$ and A-g-$C_3N_4$ surfaces. The TPD was performed on vacuum dried samples after reaction without adsorption of additional $H_2$. **c** Scheme of the suggested reaction pathway. The reduction reaction will be hindered if $H_{ads}$ binds strongly to the catalyst. **d** Reaction path for the photoconversion of nitroaromatic compounds to azoxy-, azo-aromatic compounds, and amines. NBS = Nitrosobenzene, NPH = N-phenylhydroxylamine

when using the g-$C_3N_4$ photocatalyst, whereas the A-g-$C_3N_4$ sample and pristine $TiO_2$ (Degussa Aeroxide® P25) showed much slower rates in $O_2$ depletion. We have further rationalised that such efficient reduction can be linked to the adsorption energy of the surface adsorbed hydrogen atoms ($H_{ads}$) that was generated by isopropanol oxidation during irradiation, as probed by post-mortem temperature programmed desorption (TPD, Fig. 2b and Supplementary Fig. 12). The pre-irradiated (3 h under reaction-like conditions without nitrobenzene) and subsequently vacuum dried g-$C_3N_4$ sample showed no $H_2$ desorption peak. This indicates that $H_{ads}$ is only adsorbed weakly on the catalyst surface, enabling it to react rapidly with the surface adsorbed nitrobenzene to form azo- or azoxy-benzene (Fig. 2c). In contrast, the A-g-$C_3N_4$ prepared identically showed two $H_2$ desorption peaks, suggesting a relatively strong adsorption of the photogenerated $H_{ads}$ on A-g-$C_3N_4$, which prevents the use of the $H_{ads}$ for efficient reduction reactions. The in-situ X-ray photoelectron spectroscopy analysis (XPS, Supplementary Fig. 13) also confirms that the $H_{ads}$ binds strongly on the A-g-$C_3N_4$ surface, resulting in a significant shift of the N1s peak. In contrast, the N1s peak of g-$C_3N_4$ remains unchanged upon irradiation, indicating that the $H_{ads}$ only loosely binds to the g-$C_3N_4$ surface, thus, making the catalyst ideal for the nitrobenzene reduction (Supplementary Fig. 13). In addition, the N1s peak intensity also remains unchanged, indicating that nitrogen does not leach out from the g-$C_3N_4$ catalyst. Furthermore, the g-$C_3N_4$ shows a significantly higher photocurrent compared to that of A-g-$C_3N_4$, indicating that the charge separation is also promoted by the ordered structure of g-$C_3N_4$ (Supplementary Fig. 14).

We have further explored the orgin behind the high selectivity of the nitroaromatic photoconversion into azoxy- or azo-aromatics by probing the photo reaction of the intermediates (Fig. 2d and Supplementary Table 4). Since all reactions were performed under deaerated conditions, we have ruled out the pathway involving initial reduction to aniline followed by partial oxidation to azo-/azoxy benzene. This has been also confirmed experimentally by a control reaction starting with aniline, where no N-N coupling products were formed under irradiation. According to previous mechanistic studies, nitrobenzene undergoes a gradual photoreduction process during which nitrosobenzene (NBS) and N-phenylhydroxylamine (NPH) are initially formed[27,28]. The NBS and NPH will then be further reduced to azoxy- and azo-aromatics via path (I) or to amines via path (II). The as-formed azo-aromatics may also get photoreduced into amines via path (III). We have observed that the photoreduction of NBS or of an NBS/NPH mixture (1:1 in molar ratio) results exclusively in the formation of azo- and azoxybenzene (conversion >97%) when g-$C_3N_4$ was used as photocatalyst. In contrast, the photoreduction of NBS exhibits a non-selective

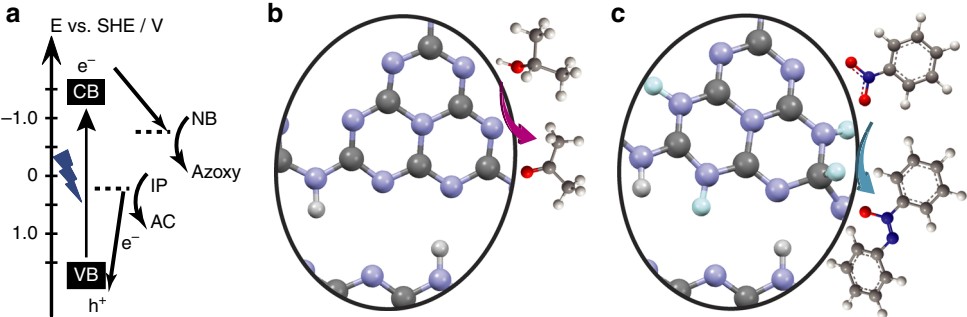

**Fig. 3** Proposed reaction mechanisms. **a** Energy scheme of the photocatalytic nitrobenzene reduction on g-$C_3N_4$. The redox potentials of nitrobenzene (NB) to azoxybenzene plus isopropanol (IP) to acetone (AC), and the CB and VB positions are referenced to the SHE[32,33]. **b** and **c** suggested possible active sites for isopropanol oxidation and nitrobenzene reduction according to calculations. Grey: C, light purple: N, white: structural H, light blue: surface adsorbed H ($H_{ads}$)

behaviour when $TiO_2$ was used as photocatalyst ([azobenzene + azoxybenzene]: amine = 44:56). The presence of NPH in the reaction (NBS: NPH = 1:1) further drives the reaction to favour the formation of the amine ([azobenzene + azoxybenzene]: amine = 18:82). Surprisingly, the photoreduction of azobenzene to amine (path III) has been found to be inactive for both g-$C_3N_4$ and $TiO_2$. These control experiments suggest that the g-$C_3N_4$ photocatalyst facilitates the N−N coupling of NBS and NPH (path I) and inhibits the reduction of NBS and NPH (path II), thus resulting in a high selectively of the nitroaromatic photoconversion into azo- and azoxy-aromatic products. Since the affinity of the aromatic molecules can be enhanced by π−π coupling effect of the tri-s-trazine unit in g-$C_3N_4$[29,30], we consider that the photogenerated NBS and NPH are co-adsorbed on the g-$C_3N_4$ surface, thus, promoting the N-N coupling for the formation of azo- and azoxycompounds (path I). In contrast, other photocatalysts (i.e., $TiO_2$) without such π−π bonding interaction possibly facilitates the reduction via path (II) to form amines.

## Discussion

The redox potentials of isopropanol/acetone and nitrobenzene/azoxybenzene, respectively, are ~0.1 and −0.8 V as referenced to the standard hydrogen electrode (SHE)[31,32]. Therefore, the photoexcited e⁻ in the conduction band (CB) of g-$C_3N_4$ is energetically favourable to initiate the nitrobenzene reduction, as depicted in Fig. 3a. Meanwhile, the isopropanol that serves as the electron donor will be oxidised into acetone, resulting in the formation of two protons and the injection of two electrons into the valence band (VB) to fill the hole ($h^+$). Since previous calculations reveal that the highest occupied molecular orbital (HOMO, equivalent to the VB) of the g-$C_3N_4$ mainly originates from the p-orbital of the two-coordinated N atoms, it is reasonable to speculate that the oxidation of isopropanol takes place on these sites (Fig. 3b)[33]. It should also be noted that the negatively charged surface of g-$C_3N_4$ under basic conditions[34] is beneficial for the deprotonation of isopropanol, thus, an improved nitrobenzene conversion is expected due to the enhanced isopropanol oxidation.

We have further explored the possible active sites for the nitrobenzene reduction reaction by employing semi-empirical MOPAC2016 calculations, to estimate the relative adsorption energies of H pairs (two H atoms) on a model crystalline g-$C_3N_4$ structure consisting of two H-bonded strips with 6 melem units (Supplementary Fig. 15). Since the nitrobenzene reduction requires removal of oxygen by two H atoms through the formation of $H_2O$, we have evaluated the relative stability of adsorbed H pairs to see if they will identify any significant sites for the reduction. In general, the H-atom pairs are found to be less stable on the crystalline g-$C_3N_4$ surface ($\Delta G_H > 0$) compared to free $H_2$, which agrees well with our post-mortem TPD results (Supplementary Table 5). The H pairs prefer to be adsorbed close to each other instead of further apart on the same melem unit, and the most stable adsorption sites are found to be the C-N bond adjacent to a bridging N atom (~ 0.5 eV) and the two-coordinated N atoms in the same ring adjacent to the bridging N atoms between melem units (~ 0.9 eV), as indicated in the Fig. 3(c). Therefore, we speculate that these possible hydrogen adsorption sites are the active sites for the nitrobenzene reduction. In addition, we found that the C-N binding configuration upon H adsorption becomes more stable when occurring on the corner melem unit, where it is only 0.1 eV less stable than free $H_2$. This could point to even more highly labile $H_2$ adsorption sites at the edges of the melem strips in real g-$C_3N_4$. A list of selected adsorption energies are provided in Supplementary Table 5.

Herein, we report on a photocatalytic strategy for controllable and efficient synthesis of a series of industrially important azo- and azoxy-aromatic compounds from their corresponding nitroaromatic precursors under visible light irradiation at RT. The g-$C_3N_4$ photocatalyst facilitates the light driven N-N coupling of the nitroaromatics, and the weakly adsorbed photogenerated $H_{ads}$ atoms contribute to the excellent catalytic performance in the multi-electron transfer process. We anticipate that the possibility of scaling-up the reaction with respect to both concentration and volume, and the use of the inexpensive g-$C_3N_4$ photocatalyst makes the reaction attractive for industrial applications.

## Methods

**Sample preparations**. The g-$C_3N_4$ photocatalyst was synthesised via conventional pyrolysis of urea at 550 °C[35], and the A-g-$C_3N_4$ sample was obtained by post heating of the g-$C_3N_4$ at 630 °C (Supplementary Fig. 1).

**Price estimation of the photocatalysts**. The prices of photocatalysts listed in Table 1 were estimated according to the catalyst compositions presented in the cited studies (Supplementary Table 2).

**Materials characterisations**. Transmission electron microscopy (TEM), $N_2$ adsorption-desorption isotherm, X-ray photoelectron spectroscopy (XPS), X-ray diffractometry (XRD), and diffuse reflectance spectrometry (DRS) were used for material characterisation (Supplementary Figs 8–11 and Supplementary Table 3). In-situ XPS and post-mortem TPD were performed to analyse the oxidation state change of N and the bond strength of the $H_{ads}$ to the catalyst surface under reaction conditions (Supplementary Figs 12 and 13). MS (Supplementary Figs 4 and 5) and gas chromatography (GC) were used to investigate the reaction mechanism.

**Photocatalytic process analysis**. A home-built vacuum-gas line was used to perform the photocatalytic reactions (Supplementary Figs 2 and 3). GC was used to determine the concentrations of the reactants and the products.

**Calculations**. The semi-empirical molecular orbital geometry optimisations were performed in MOPAC2016, all using the PM7 Hamiltonian (Supplementary Fig. 15 and Supplementary Table 5)[36]. All details can be found in the Supplementary Methods and Supplementary Discussion.

**Data availability**. The data that support the findings of this study are available from the corresponding author upon request.

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

## Acknowledgements

R.S. would like to thank the NSFC for financial support (projects number: 21503257 and 21601198). Y.D. would like to thank the Sino-Danish Centre (SDC) for the Ph.D Scholarship. N.L. acknowledges the Villum Foundation Young Investigator Programme (VKR023449) and the Danish National Research Foundation (Carbon Dioxide Activation Centre, DNRF 118). We acknowledge the support from the Carlsberg Foundation, and we would like to thank Dr. Yi Wang, Dr. Yifeng Yun, and Ms. Yiling Bai from Synfuels China Technology Co. Ltd. for the TEM and TPD analysis. We thank Dr. Chao Wang from Shenzhen University for providing space to perform the upscaled reaction.

## Author contributions

Y.D., R.S., and N.L. conceived the studies. Y.D., C.L., and Y.S. synthesised the samples. Y.D. and C.L. performed the photocatalytic tests. Y.D., C.L., N.L., and R.S. carried out the material characterisation. T.L. performed the calculations. J.X., Y.L., H.N., and F.B. were involved in the design of the experiments, the discussion of the results, and the writing up of the manuscript.

## Additional information

**Competing interests:** The authors declare no competing financial interests.

