## [Peer Review File · Nature Communications]

Editorial Note: This paper has been previously reviewed at a journal that does not have Transparent Peer Review. Below are the reviewer comments and author responses for this paper while at Nature Communications.

Reviewers' comments:

Reviewer #1 (Remarks to the Author):

The authors synthesized inexpensive graphitic C₃N₄ (g-C₃N₄) catalyst for synthesising a series of important azo- and

azoxy- aromatic derivatives from their corresponding nitro-aromatics with very high selectivity. The results obtained are

very interesting and might also have important implications. Therefore, I think the manuscript could be accepted for

publication after the following minor issues are addressed:

(1) It is suggested that the mechanism pictures for photocatalysis over g-C₃N₄ particles should be provided, which should

including the excitation of g-C₃N₄ from its VB to CB, charge transfer and surface reaction. Fig. S15 is too simple.

(2) Is the negatively charged surface of g-C₃N₄ advantageous for enhancing the selectivity?

(3) If possible, the exact intermediate products could be identified by in-situ FTIR or other measurements.

(4) The author should carefully reveal the exact active sites for in g-C₃N₄ the photocatalysis.

(5) More recently published papers can be referenced:

Nat. Commun., 2013, 4, 2681; Nat. Commun., 2015, 6, 7258; Nat. Commun., 2016, 7, 12165; Appl. Surf. Sci., 2017, 391, 72-

123; J. Mater. Chem. A, 2015, 3, 2485-2534; Adv. Mater., 2015, 27, 2150-2176.

Reviewer #2 (Remarks to the Author):

The authors have followed the recommendations of the reviewers.

They have extended the scope and have optimized the reaction conditions, in all the cases, except one, the conversion now exceeds 90%. They have also performed two photocatalytic syntheses of asymmetric azobenzene and preliminary results show a selectivity of ~50%.

Since as the authors mention that the conditions and reactor design have not been optimized and the scaled-up tests were only included in the study to demonstrate its industrial potential, the word "scalable" from the title should be removed.

After that, I recommend its publication in Nature Communications.

Response to referee 1

The authors synthesized inexpensive graphitic C₃N₄ (g-C₃N₄) catalyst for synthesising a series of important azo- and azoxy- aromatic derivatives from their corresponding nitro-aromatics with very high selectivity. The results obtained are very interesting and might also have important implications. Therefore, I think the manuscript could be accepted for publication after the following minor issues are addressed:

1. It is suggested that the mechanism pictures for photocatalysis over g-C₃N₄ particles should be provided, which should including the excitation of g-C₃N₄ from its VB to CB, charge transfer and surface reaction. Fig. S15 is too simple.

Response:

We agree with the referee that the schematic mechanism shown in the SI (Fig. S15) could be improved by including the excitation, charge transfer and surface reaction process.

We have added Figure 3 in the revised manuscript with additional discussion and references in the "Discussion" section on page 5.

2. Is the negatively charged surface of g-C₃N₄ advantageous for enhancing the selectivity?

Response:

The referee raised a very interesting comment. The g-C₃N₄ used in our manuscript is synthesized from pyrolysis of urea at 550 °C, which is characterized by a point of zero charge (pzc) of ~ 5 according to the zeta potential analysis (see Zhu *et al.*, *Appl. Surf. Sci.* 2015, 344, 188-195). Therefore the referee is absolutely correct that the surface of the g-C₃N₄ is negatively charged.

However, we consider that the negatively charged surface g-C₃N₄ will mainly influence the reactivity. At neutral pH, the selectivity to azoxybenzene remains to be ~ 100%, but the conversion has dropped to 4% under identical irradiation conditions (450 nm 0.03 W·cm⁻²). This suggests the negatively charged surface promotes the deprotonation of isopropanol, thus providing more surface adsorbed hydrogen for the nitrobenzene photo-reduction.

Our control experiments indicate that the negatively charged surface of g-C₃N₄ is unlikely to influence the selectivity of the nitrobenzene photo-reduction. The possible intermediates in photocatalytic nitrobenzene reduction reaction are nitrosobenzene (NBS) and N-phenylhydroxylamine (NPH) according to an electron spin resonance study (see Maldotti *et al.*, *J. Photochem. Photobiol. A*, 2000, 133, 129-133), as shown in the following reaction equations:

Obviously the formation rate of the secondary amine radical (marked in the dashed square) and the coupling rate of NBS and NPH is the key to tune the selective to amine or azoxy/azo aromatics. Our control experiments (see Table S4 in the SI) show that the $g\text{-C}_3\text{N}_4$ photocatalyst accelerated the coupling reaction of NBS and NPH and inhibited the further reduction of NPH to aniline. In contrast, the poor selectivity of P25 to azo-/azoxy-benzene originates from the fast conversion of NPH to the amine radical, and a slow coupling rate.

We have revised the manuscript on page 5 accordingly.

3. If possible, the exact Intermediate products could be identified by *in-situ* FTIR or other measurements.

Response:

Identification of the exact intermediates is of great importance to understand the reaction mechanisms and to help the design of high performance photocatalysts. We have tried using *in-situ* FTIR under semi-reaction conditions, however the intermediates are too reactive to be followed by FTIR. We have also performed electron spin resonance (ESR) analysis of the irradiated catalyst-reactant suspension at room temperature but observed no paramagnetic signals. We have also tried irradiating the $g\text{-C}_3\text{N}_4$ with pre-adsorbed nitrobenzene at liquid nitrogen temperature, but detected no signals. This is probably due to the reactants freezing under these conditions.

However, as presented in the aforementioned equations (Eq. 1 and 2), although identifying the intermediates are challenging, we have performed control experiments to elucidate possible intermediates. It has been confirmed by several reports that the NBS and NPH are the relatively stable intermediates of the nitrobenzene reduction (see Maldotti *et al.*, *J. Photochem. Photobiol. A*, 2000, 133, 129-133; Brezová, *et al.*, *J. Photochem. Photobiol. A* 2003, 155, 179-198). Our control experiments (see Table S4 in the original SI) also show that the photo-reduction of NBS and NBS + NPH both result in the formation of azoxy-/azo-benzene (sel. > 97%), indicating both NBS and NPH are the reaction intermediates in the photocatalytic nitrobenzene reduction over $g\text{-C}_3\text{N}_4$. It should also be noted that azobenzene cannot be further photoconverted into amine, neither by using $g\text{-C}_3\text{N}_4$ nor TiO_2 , indicating amine must come from the reduction of the secondary amine radical (Eq. 1). The low selectivity to amine by $g\text{-C}_3\text{N}_4$ therefore suggests that the secondary amine radical is unlikely to be a reaction intermediate.

In a follow-up project we will aim at using time resolved electron spin resonance to probe the exact intermediates *in situ* to complete this study.

We have revised the manuscript on page 4 with two additional references accordingly.

4. The author should carefully reveal the exact active sites for in *g-C₃N₄* the photocatalysis.

Response:

We thank the referee for this constructive comment. Previous calculations have shown that the HOMO (or VB) is mainly originated from the p orbital of the two-coordinated N atoms in the melem ring, whereas the hybridized N p and C p orbitals contributes to the LUMO (or CB) (see Lau *et al.*, J. Am. Chem. Soc. 2015, 137, 1064-1072). Therefore it is reasonable to speculate the oxidation reactions take place at the two-coordinated N atoms.

In our case, an electron from the N p orbital will be excited to the hybridized N p and C p orbital and creates a hole at the N p orbital upon irradiation. Then the isopropanol will be oxidized to acetone at the two-coordinated N atoms and injects an electron to fill the hole of the N p orbitals. We have now performed semi-empirical molecular orbital calculations to estimate the adsorption sites of hydrogen atoms on *g-C₃N₄*. In general the H atoms are thermally unstable compared to free H₂ on the crystalline *g-C₃N₄* surface ($\Delta G_H > 0$), which agrees well with our post-mortem TPD results. The two relatively stable sites are found to be the atoms of the C-N bond adjacent to the bridging N atoms (0.5 eV) and the two 2-coordinated N atoms in the same ring adjacent to the bridging N atoms between melem units (0.9 eV), as indicated in the new Fig. 3(c). A further exceptionally stable version of the C-N bonding site (~ 0.1 eV) for H atom pairs, found only on the edge melem unit, also points to the possibility of highly labile H₂ adsorption edge sites in crystalline *g-C₃N₄*. Therefore we propose that these sites are the possible active sites for nitrobenzene reduction.

We have revised the manuscript on page 5 and the SI with additional calculation details accordingly.

5. More recently published papers can be referenced: *Nat. Commun.*, 2013, 4, 2681; *Nat. Commun.*, 2015, 6, 7258; *Nat. Commun.*, 2016, 7, 12165; *Appl. Surf. Sci.*, 2017, 391, 72-123; *J. Mater. Chem. A*, 2015, 3, 2485-2534; *Adv. Mater.*, 2015, 27, 2150-2176.

Response:

We thank referee 1 for the suggested references that are relevant to the design and application of the *g-C₃N₄* based materials. We have cited these references in the revised draft with a brief summarization of *g-C₃N₄* (page 1).

Response to referee 2

The authors have followed the recommendations of the reviewers.

They have extended the scope and have optimized the reaction conditions, in all the cases, except one, the conversion now exceeds 90%. They have also performed two photocatalytic syntheses of asymmetric azobenzene and preliminary results show a selectivity of ~50%.

Since as the authors mention that the conditions and reactor design have not been optimized and the scaled-up tests were only included in the study to demonstrate its industrial potential, the word "scalable" from the title should be removed.

After that, I recommend its publication in Nature Communications.

Response:

We thank referee 2 for the positive comments. We agree that the scaled-up tests only suggest the industrial potential, and that the reactor design has not been optimized. Therefore we follow the suggestion by removing the word "scalable" in the title.

REVIEWERS' COMMENTS:

Reviewer #1 (Remarks to the Author):

It can be accepted as it is!